# Exploring the Impact of a Family-Focused, Gender-Transformative Intervention on Adolescent Girls’ Well-Being in a Humanitarian Context

**DOI:** 10.3390/ijerph192215357

**Published:** 2022-11-21

**Authors:** Ilana Seff, Andrea Koris, Monica Giuffrida, Reine Ibala, Kristine Anderson, Hana Shalouf, Julianne Deitch, Lindsay Stark

**Affiliations:** 1Brown School of Social Work, Washington University in St. Louis, St. Louis, MO 63130, USA; 2Independent Researcher, Capetown 8001, South Africa; 3Women’s Refugee Commission, New York, NY 10019, USA; 4Weill Cornell Medical College, New York, NY 10065, USA; 5Mercy Corps Headquarters, Portland, OR 97204, USA; 6Mercy Corps Jordan, Building No. 8, Tabasheer 3 Street, 7th Circle, Amman, Jordan

**Keywords:** adolescent girls, mental health, well-being, refugees, gender-transformative, evaluation, family functioning

## Abstract

While family functioning interventions show promise for improving adolescent girls’ well-being in humanitarian contexts, few programs employ a gender-transformative approach to maximize benefits for adolescent girls. This paper presents findings from a mixed-methods pilot evaluation of a whole-family, gender-transformative intervention conducted with Syrian refugee families in Jordan. The Siblings Support of Adolescent Girls in Emergencies program was implemented with 60 Syrian refugee households in Azraq and Za’atari camps in Jordan. A quantitative survey was administered to 18 households at baseline and endline, and researchers conducted qualitative interviews and focus group discussions with caregivers, paired interviews and participatory discussions with adolescents, and key informant interviews with program mentors. Paired *t*-tests revealed statistically significant improvements in mental distress, resilience, and gender equitable attitudes in the full sample and for girls only and marginally significant improvements in family functioning. Qualitative findings revealed improvements in four domains of girls’ well-being—self-efficacy, self-confidence, pro-social behavior, and mental health—through three primary pathways: family members’ increased gender equitable attitudes, healthier intrahousehold communication, and greater affective involvement. Findings from this mixed-methods evaluation point to the potential value in merging gender-transformative and whole-family approaches in humanitarian programming to maximize positive impacts for adolescent girls.

## 1. Introduction

The impact of persisting forced displacement on growing refugee populations poses a pressing global health challenge [1,2,3]. Despite underestimation of diagnoses due to stigma and difficulties with evaluation in humanitarian settings, the literature concurs that refugees, particularly forcibly displaced minors, face increased risks for mental illness and distress [1,3,4]. Women and girls exhibit an increased likelihood of mental distress, with worse outcomes observed among minors [3,5,6,7].

Depression, anxiety, self-harm, and psychosomatic symptoms constitute the most common forms of mental illness among migrant girls [3,8,9]. Cyranowski and colleagues’ theory of adolescent gender differences in lifetime rates of major depression posits that the stressors faced by adolescent girls are at least partially a result of the adolescent period of development marked by hormonal changes during puberty; however, intensified gender socialization, stressful life events, and impaired adolescent transitions also heighten their vulnerability to mental health disorders [10]. Further, while migration itself, along with exposure to conflict, loss of family members, child labor, and other life-threatening experiences are recognized predisposing factors for mental distress in refugee youth, adolescent girls face additional gendered exposures [3,8,11]. In humanitarian settings, girls’ heightened risk of exploitation, sexual violence and torture, early marriage, removal from school, human trafficking, and other forms of gender-based violence make them more prone to developing mental health sequelae [3,8]. In fact, analyses of ecological risk factors for mental health in refugee women and girls highlighted intimate partner violence and conflict-related violence as factors with the greatest odds of increasing mental health disorders [11,12].

Gender inequitable norms that define acceptable and expected behaviors for men and women—often in ways that marginalize women and girls—can be heightened in a humanitarian setting and have a well-documented negative impact on the mental well-being of adolescent girls [13,14,15]. Findings from studies such as Gender and Adolescence: Global Evidence (GAGE), conducted in Ethiopia and Bangladesh, demonstrate a strong association between gender inequitable community norms and reduced self-esteem and psychological well-being, particularly in adolescent girls [13]. Often limiting girls’ mobility, prioritizing domestic responsibilities over education, and limiting freedoms after the onset of puberty, gender inequitable attitudes play a role in diminishing girls’ sense of self-efficacy and hope [13]. Current evidence indicates that pressured gender conformity and internalization of social norms devaluing women adversely impact girls’ self-esteem and, subsequently, this impaired resilience results in their mental and emotional distress [13,16,17]. This relationship between community- and family-held gender inequitable attitudes, and girls’ decreased self-esteem, is prevalent in refugee contexts, as supported by analysis of Sudanese refugees in Ethiopia [15].

The significant financial strain often experienced by families in conflict and post-conflict settings may also negatively impact the well-being of adolescent girls. As employment opportunities for men and boys are often more limited in emergencies, women and girls often take on income-generating responsibilities; moreover, in cultures that prioritize men’s roles as breadwinners, such changes in intrahousehold gender dynamics can threaten the status quo in patriarchal societies and exacerbate gender-based tension within families [18,19,20,21]. As adolescent girls are positioned at the bottom of both age and gender hierarchies, they may be especially susceptible to behavioral control by and punishment from their fathers and brothers [22]. For example, in unstable humanitarian contexts, regard for inequitable gender norms as promoting girls’ safety often results in greater use of controlling and punitive measures with adolescent girls, which may in turn impact their well-being [23,24]. Further, the acute and daily stressors germane to emergencies can increase parents’ reliance on unhealthy parenting behaviors as a coping mechanism, further threatening the well-being of adolescent girls [25,26].

In line with findings on the impact of intrahousehold gender inequity on girls’ well-being, evidence from humanitarian settings also highlights the protective role healthy family relationships can play in fostering girls’ mental health. In conflict and post-conflict zones at large, caregivers’ psychological health, family cohesion, and peer support proved vital in fostering resilience in affected children [4,23,27]. Adolescents in Gaza expressed reduced anti-social behavior, depressive sequelae, and aggression in households characterized by increased parental support and warm parenting practices [27,28]. These findings suggest the potential of familial interventional targets for bolstering resilience and psychological well-being in girls in humanitarian contexts.

Though family-based approaches have been deemed effective in the treatment and prevention of distinct mental disorders in youth in psychological practice in high income countries, evidence supporting the generalizability of these impacts in humanitarian settings remains limited and mixed [29]. In a recent systematic review of caregiver interventions in forcibly displaced communities, parenting programs demonstrated little to no impact on children’s psychological well-being, including internalizing and externalizing behaviors [30]. Conversely, Syrian parents who participated in War Child Holland’s Caregiver Support Intervention reported improved psychosocial health in their children, while the children of parents who participated in a teaching recovery techniques parenting program exhibited decreased post-traumatic stress [31,32]. Ultimately, few evaluations exist which assess the role of family functioning on girls’ psychosocial well-being in humanitarian contexts, especially those involving men and boys. Further, to date, a dearth of data exists analyzing these relationships at the household level, engaging both sibling and caregiver dynamics.

Nascent evidence from a family-focused, gender-transformative intervention in northeast Nigeria, the Sibling Support for Adolescent Girls in Emergencies (SSAGE), demonstrated improved pathways of communication between male and female caregivers and adolescents—leading to increased positive involvement and parental warmth toward adolescent girls [33]. However, additional research is needed to understand the intervention’s follow-on impacts for adolescent girls’ own well-being and mental health. This paper presents findings from a recent mixed-methods pilot evaluation of SSAGE, conducted with Syrian families in Jordan. Results from this study help to expand the evidence base on interventions addressing the well-being of adolescent girls in humanitarian settings—specifically whole family programming targeting caregiver and male sibling engagement.

## 2. Methods

### 2.1. Setting

This study took place within two of the three refugee camps in Jordan hosting Syrian refugees: Za’atari and Azraq [34]. Since 2014, approximately 36,000 Syrian refugees have been relocated to these two camps, where the majority of residents are children and live in female-headed households. Refugees living in Za’atari and Azraq face extreme restrictions on their movement and cannot leave the camps without obtaining permission from the Jordanian government [35]. Already exposed to war and related trauma in their country of origin, Syrian refugees in Jordan face further challenges, including insecure housing, limited access to education and healthcare, and minimal employment opportunities. Further, restrictive gender norms work to marginalize women and girls and put them at risk of experiencing interpersonal violence and harassment both within and outside the home [36]. However, culture and underlying gender norms are not static, and there is heterogeneity in the behaviors expected of adolescent girls in this setting—with gender norms influenced to varying degrees by families’ religious, socioeconomic, and family identities, as well as many other dimensions of identity [37,38]. Further, the disruption to the status quo that often accompanies forced displacement creates opportunities for gender transformative programming to transform adolescent girls’ position in society.

### 2.2. Intervention

The SSAGE intervention was implemented with 60 Syrian refugee households living within the Azraq and Za’atari camps in northern Jordan in two cycles: the first from June to September 2021 and the second from October 2021 to January 2022. Thirty households from each camp were identified and selected for participation primarily through Mercy Corps’ existing ISHRAK project, a youth and family support program that provides psychosocial support to fortify bonds between family members. Eligible families included those with a male and female caregiver, an adolescent girl aged 10–14, and an older male sibling aged 15–19 living at home. The four members of each participating household attended program sessions with their gender and age group over the course of twelve weeks, which were led by program mentors who were also Syrians residing in the Azraq and Za’atari camps. Trained by Mercy Corps, these program mentors facilitated four distinct but synchronous curricula that were designed to stimulate self-reflection and discussion amongst participants on topics related to gender, power, violence, interpersonal communication, and healthy relationships (please see [39] for more information on the SSAGE intervention, including the curricula topics). Prior to the implementation of the program, the SSAGE curricula were contextualized via a community consultation process to ensure the content and facilitation methods were appropriate and relevant to the Syrian refugee communities in the Za’atari and Azraq camps.

A mixed-methods pilot evaluation of the SSAGE intervention was conducted to better understand the program’s impact on participating adolescent girls’ well-being. Evaluation activities included administration of a baseline and endline survey questionnaire with approximately half of the participating households during cycle 2, as well as in-depth interviews, focus group discussions, participatory activities, and key informant interviews both for cycle 1 and cycle 2.

### 2.3. Quantitative Methods

A survey questionnaire was administered at baseline and within two weeks post-intervention (endline) to assess changes in outcomes of interest over the course of the intervention. Eighteen families (nine from each camp) were randomly selected to participate in the quantitative evaluation. At least 1 family member from all 18 households agreed to participate in this component of the study; baseline data were collected for 68 participants, as 4 individuals did not participate in data collection. Written informed consent was obtained from all study participants prior to data collection, and all surveys were enumerator-administered in private spaces using the Kobo platform.

The survey questionnaire included several previously validated measures to capture outcomes of interest on mental distress, resilience, gender equity, and family functioning; all outcomes were self-reported. Mental distress was measured using the Kessler-6 scale (Cronbach’s alpha in the full sample = 0.77), which has been previously used with Syrian refugees in Lebanon [40,41]. Respondents are prompted to indicate the frequency, in the last 30 days, at which they felt nervous, hopeless, restless or fidgety, depressed, that everything was an effort, and worthless; the measure was operationalized such that higher scores signal less distress. The Child and Youth Resilience Measure-12 (CYRM-12), which has been validated in Arabic with Syrian refugees in Jordan, was used to measure resilience among adolescent boys and girls (Cronbach’s alpha = 0.63) [42,43]. Respondents are asked to share their extent of agreement (on a 5-point Likert scale, from ‘Not at all’ to ‘A lot’) with twelve items, such as “I try to finish things I start”, “I know where to go to ask for help”, and “I am aware of my own points of strength”, among others. Responses across items are summed, whereby higher responses represent greater resilience. Items to construct the measure of gender equitable attitudes included a subset of items employed in an Arabic version of the Gender Equitable Men’s scale in Egypt (Cronbach’s alpha = 0.73). Nine items were used, capturing a range of attitudes toward gender equity, such as “A woman’s most important role is to take care of the home and cook for the family”, “To be a man, you need to be tough”, and “A man should have the final word about decisions in the home”. Respondents provided their level of agreement on a 4-point Likert scale. The final score was created by averaging the scores across the nine items, with higher scores representing more gender inequitable attitudes. Finally, family functioning was measured using the Family Attachment and Changeability Index (Cronbach’s alpha = 0.67) [44]. Sixteen items were used to assess dimensions of family functioning relating to discipline, adaptability, communication, inclusion, and others; the final measure was constructed such that higher values signal greater family functioning.

Quantitative data were collected by a mixed-gender Jordanian research team experienced in social protection research in the Azraq and Za’atari camp communities. A research team from Washington University in St. Louis provided remote training for data collectors prior to data collection. Descriptive statistics were calculated for the full sample and each cohort (adolescent girls, adolescent boys, female caregiver, and male caregivers). Given that the intervention was ultimately designed to improve well-being for adolescent girls and improve psychosocial outcomes for the other cohorts, we examined changes in the outcomes of interest for both adolescent girls and the full sample. Differences between outcomes at baseline and endline were assessed using paired *t*-tests. Statistical analyses were carried out by a Washington University researcher using Stata16.

### 2.4. Qualitative Methods

Qualitative data were collected one month after the completion of each program cycle in order to explore any perceived changes related to gender equity, family functioning, violence against women and girls, and girls’ experience of well-being following the program. Qualitative participants were recruited using a criterion sampling based on the following factors: gender, age, and program attendance rate. In total, 137 participants were selected from the first program cycle and 76 were from the second program cycle (see Table 1 for a breakdown of participants). Qualitative methods included in-depth interviews (IDIs) and focus group discussions (FGDs) with adult caregivers, as well as paired interviews and participatory group discussions for adolescents. Adolescent research was facilitated using youth-friendly methods, including story completion or arts-based activities. The group-based research activities were structured around vignettes which depicted scenarios of family and peer-based interactions; this approach gave rise to participant discussion and debate about issues related to gender norms, gender inequity, women’s rights, and resource scarcity in their own communities. Group activities were facilitated by two data collectors, one of whom documented nonverbal communication and interactions, so that the full scope of the groups’ agreement and debate on various issues was documented.

In contrast to the group-based research activities, the IDIs and paired interviews yielded more discussion from participants about the impact of these issues in their own lives and the lives of their family members. Additionally, eight SSAGE program staff, including mentors, were purposely selected for key informant interviews (KIIs), during which they discussed their perceptions of the acceptability and impact of the program.

Qualitative data were collected by the same mixed-gender Jordanian research team that collected the quantitative data. Prior to the start of each data collection round, the research team undertook a review training focused on qualitative interviewing, research ethics, and use of Mercy Corps’ Community Accountability Reporting Mechanisms. Research assistants transcribed audio recordings into Arabic at the conclusion of each interview or discussion. An external translator then translated the Arabic transcripts into English. Another Arabic-speaking member of the study team conducted a quality control review by spot-checking transcripts against audio files to ensure the transcription and translations accurately represented participants’ thoughts and opinions.

After data collection concluded, an analysis team from Mercy Corps Jordan, Women’s Refugee Commission, and Washington University in St. Louis participated in a workshop during which they reviewed selections of data and used deductive and inductive methods to develop a preliminary codebook. The analysis team then iteratively adapted the codebook over several months, piloting it on various subsections of the data and convening to discuss suggested adaptations to the code framework. Once a final version of the codebook was reached, the lead qualitative researcher established a coding consensus with six other members of the analysis team, who then proceeded to code the remainder of the qualitative data. They then analyzed the data using Dedoose qualitative data analysis software.

All study procedures were approved by the Health Media Lab institutional review board (HML #983WRCO21).

## 3. Results

### 3.1. Quantitative Results

Baseline and endline data were collected from 68 participants across 18 families. Table 2 presents basic demographic information for the full sample as well as each participant type. The average age of adolescent girl participants was 12.5 years. All adolescent girl participants were currently attending school, and only one adolescent girl in the sample had ever worked for pay.

Paired *t*-tests revealed improvements across all four outcomes of interest in the full sample between baseline and endline (see Table 3). Specifically, the Kessler scale (where a higher score signals less distress) increased from 13.67 at baseline to 15.85 at endline (*p* = 0.007), the gender equity score (where a higher score represents more gender equitable attitudes) increased from 2.34 to 2.49 (*p* = 0.004), and the family functioning score rose from 61.33 to 64.69 (*p* = 0.044). Resilience, which was measured among children only, also increased from 39.83 to 42.62 (*p* = 0.03).

Figure 1 depicts changes in outcomes of interest between baseline and endline for adolescent girls only. Although a small sample, similar improvements were observed for this sub-group of participants, with statistically significant improvements in mental distress, resilience, and gender equitable attitudes and marginally significant improvements in family functioning.

### 3.2. Qualitative Results

The qualitative arm of the study allowed for further exploration of the improvement in girls’ well-being, providing insight into the ways in which participants conceptualized these changes. Participating adolescent girls and their family members highlighted improvements in four major domains of girls’ well-being following their participation in the SSAGE sessions: self-efficacy, self-confidence, pro-social behavior, and mental health. These domains were impacted by distinct pathways, including improvements in gender equitable attitudes and awareness around girls’ safety and well-being among family members; changes in control of and communication around girls’ mobility and behaviors; and family members’ affective involvement/emotional connection with adolescent girls. The qualitative analysis first examines the four domains of adolescent girls’ well-being and then explores the pathways that led to these reported improvements in this well-being.

#### 3.2.1. Adolescent Girls’ Well-Being

##### Self-Confidence

Many adolescent girls discussed how the skills they learned and developed in the SSAGE program increased their confidence when moving around the camp and helped them to define and assert their boundaries within social interactions. As the program equipped girls with increased knowledge on how to protect themselves and how to ask for help in dangerous situations, girls became more confident in their ability to go to the market or visit friends. Many girls described feeling “stronger” and “less afraid” after attending the program. For example, one adolescent girl noted, “I used to be afraid, afraid of walking in the street alone or to go to the market to buy something by myself. Now it’s ok, I walk, and I have confidence and I don’t let anyone see me as a weak person so that they don’t take advantage of me. I strengthen myself so that they don’t abuse me” (Za’atari-PI-AG-13022022(1)(E)(1)).

Girls further developed their self-confidence by being encouraged to share their opinions and being taught to be firm in their boundaries. One girl said, “… before participating in this program, I did not know how to express my opinion, whatever you ask me I stayed silent, but now anyone asks me I express my opinion and share what I have” (Azraq-PI-AG-18102021(3)). Some adolescent girls specifically noted how their increased confidence enabled them to improve the way they communicate with their family. For example, one girl said, “I have changed. I talk to my brother and my dad openly and I have self-confidence, I just talk to them normally. I used to stutter and don’t know what to say even if I was sure of the topic and repeated it 100 times in the kitchen, I didn’t know what to say. Now, when I need anything, I just go to my dad and my brother and my mom and tell them what I want clearly” (Azraq-PI-AG-14022022 (3)(E)). Other girls expressed similar feelings, sharing that they felt more comfortable in sharing their views with family members and that, in turn, their family members were more open to listening to their adolescent girls’ opinions.

Adolescent girls also learned how to identify and respond to unsafe social interactions among their peers, including bullying. By talking about bullying in the SSAGE program, girls became more adept at identifying peer-to-peer bullying among adolescent girls and addressing bullying outside of the confines of SSAGE programming. Some girls passed along what they learned about bullying to members of the community, demonstrating they felt confident in expressing their opinions and their boundaries in unsafe peer-to-peer social interactions. For example, one mother said, “I saw girls becoming more active with the program. They got support that helped them become stronger and more confident. Facilitators and other girls in their sessions helped them become more confident… My daughter talked to my neighbor about bullying. She now knows what bullying is. She knows right from wrong. She would tell people that something involves bullying and that bullying is wrong” (Azraq-FGD-CF-17102021).

##### Self-Efficacy

Overall, participants expressed that the SSAGE program had positive impacts on adolescent girls’ sense of self-efficacy, particularly with regards to protecting themselves, communicating their ideas, and engaging in healthy social interactions. Participants described seeing changes to the ways that girls were able to handle problems, defend themselves, and deal with risky situations and challenging tasks. One program mentor shared a story about a girl who experienced abuse and violence at home. While the participant still appeared to experience violence perpetrated by her brother, the SSAGE program seems to have positively impacted her ability to defend herself from this violence:


*“A mother told us that her daughter told her that her brother hit her, rebuked her, and bossed her around all day long. After a while, the mother told me that ‘my daughter benefited from you a lot and she was affected by your personalities a lot. Now when her brother hits her, she will defend herself, because of the information she gets from Mercy Corps.’ I myself learned and gained information and benefited from the teacher a lot”. She defends herself now and she says: “it’s not only you who got information, I did too”.*
(KII02)

Girls also discussed being able to identify specific ways to protect themselves outside of the home and where to go if an incident occurs, for example, “… if someone crosses our way or says something we should report to a woman or a girl (note: a service center) where they listen to us and we can make a complaint. If someone crosses our way, they tell us not to talk to him because he won’t stop bothering us if we do. If he throws a letter, we shouldn’t take it. We should just ignore him. Tell your mother and your brothers whom you trust. Tell them so that they go and tell him to stay away from you” (Azraq-PAR-AG-17102021(1)). Caregivers of adolescent girls also noted that their daughters learned how to better protect themselves around the community after participating in the program. One mother said,

*“They taught kids how to be strong and defend themselves when they go out alone, without us. They didn’t know how to, maybe we didn’t give them clear information. Now they can react and defend themselves. A girl knows how to defend her education and future now. Many other things… I wanted my daughter to be stronger. If the program was given again, I would attend the sessions because my daughter benefited. I love to see my daughter strong”*.(Azraq-FGD-CF-15022022 (E))

Many girls also expressed that they learned about their rights and “learned how to speak” during the program, developing greater self-efficacy in communicating their wants and needs. For example, one adolescent girl shared, “I couldn’t speak my mind before the program. Now I can say my opinion to my parents and brothers, and they listen to me and act upon it” (Azraq-PI-AG-18102021(1)).

##### Prosocial Behavior

Partly as a result of feeling more confident expressing their feelings, opinions, and boundaries, participating adolescent girls engaged in more prosocial behaviors with their peers and family members. Family members noticed differences in how their daughters interacted with the entire family after participating in the SSAGE program. Many caregivers said their daughters began to share more about their lives, listened better, and spent more time with the family. In one interview, a father of an adolescent girl said, “My daughter participated in the course and started to become more responsive with us. She didn’t respond to us easily before. Now she comes back from the session and tells us what she took, what the title of the session was and the questions it included. She is more interactive with her mom now” (Za’atari-IDI-CM-24102021(3)). These observations by family members mirror statements girls made about how they felt like they cooperate more with their siblings and parents after the SSAGE program.

Caregivers also noted that adolescent girls seemed to be more comfortable sharing dimensions of their personalities that they had not revealed to family members prior to the program, enabling girls to develop stronger relationships with their family members. One mother described how her daughter opened up after participating in the program: “My daughter used to be an introvert. She used to spend the whole day in her room. She did not join us or talk to us. She has changed, mashallah (God has willed it). She sits with us and jokes with us. I never knew that my daughter has such a sense of humor! My daughter has changed after the sessions” Azraq-IDI-CF-18102021(1).

With respect to peer-to-peer interactions, girls learned how to better cooperate with their friends following SSAGE participation. Lessons on bullying also reportedly promoted prosocial behaviors among girl participants. In one interview, a girl said, “I fought with girls wherever I went. The sessions gave us awareness. I stopped fighting with girls” (Azraq-PI-AG-14022022(4)(E)). During a group activity asking girls to tell stories about their experience in the program, one group said that girls felt more respected and experienced less bullying by their female peers, stating, “They used to talk behind her back and used to call her names…Her friends respect her more when they see her now. They used to laugh at her” (Za’atari-PAR-AG-20102021). In some cases, girls’ increased self-confidence led to them interacting more with their peers overall. In other instances, some girls talked about how they did not have friends in the community until they met friends through the SSAGE program.

##### Improved Mental Health

Some participants discussed how they believed adolescent girls’ mental health improved as a result of the SSAGE program. In these instances, girls and their family members attributed adolescent girls’ improved mental health to other outcomes of the program, such as increased self-confidence or prosocial behaviors. In one participatory session, adolescent girls were asked to create stories illustrating the ways in which participating in the program impacted their lives. One such story demonstrated how the program helped them improve their confidence and communication:


*“There was a desperate girl who didn’t feel strong, felt tired most of the time, and spent most of the day alone in her room. She didn’t mingle with people and didn’t sit with her family. She only left her room to eat. She didn’t have anything to talk about with her parents. When Mercy Corps visited the family and told them about the Siblings Program, they agreed to participate in the sessions. They started to attend sessions about self-confidence and self-defense. She started to learn about how to strengthen herself and how to communicate. After the sessions, her life started to become normal, and she started to talk to her parents and speak up her mind. She now tells her family about any problem she faces. She talks to her mom, dad, and brother”.*
(Za’atari-PAR-AG-20102021(1))

This story reveals how, prior to the SSAGE program, some girls may have experienced symptoms of poor mental health and felt unable to clearly and productively communicate with their families. Through the confidence they gained and the skills they learned in the program, girls began to feel emboldened and able to express themselves. By improving their capacity to express their feelings, adolescent girls were also better able to ask their families for help. As such, girls could build a support system to help them through difficult problems. Adolescent girl participants shared that feeling better understood and supported helped to ease their distress.

In some cases, girls’ parents also noted how their daughters’ increased pro-social behaviors contributed to improvements in mental health. For example, one mother shared, “As for me, my daughter faced depression. She is lonely and doesn’t have any friends. She used to have friends here, but they left. She remained alone at home and doesn’t want to go here and there. I would tell her ‘Go see the neighbor’s girls at the center…’; she would instantly say ‘no.’ Now she visits the neighbor’s girls, thank God, and she is happy. She found girls to talk to, have fun and laugh. Thank God, she got rid of depression” (Azraq-FGD-CF-17102021). Other adolescent girls talked about how they developed more awareness of their emotions and a better understanding of how to handle difficult feelings. One girl said, “… there was so many things I did not know and now I know because of the program, like how to get rid of anger for example, if she for example gets angry fast, to stop this anger, and if she was afraid or sad there is also a way to get rid of sadness, there is no negative impact on her” (Azraq-PI-AG-18102021(3)). These newer coping skills helped girls to navigate difficult situations and handle challenging relationships, both of which fostered improved senses of well-being.

#### 3.2.2. Improvements in Family Members’ Gender Equitable Attitudes and Awareness around Girls’ Safety and Wellbeing among Family Members

SSAGE participants generally discussed a greater awareness and scrutiny of gender inequitable norms, including rigid conceptions of masculinity, gender roles, girls’ education, and early marriage, as well as an improved understanding of safety within the community for girls. Changes in family members’ gender inequitable attitudes reportedly indirectly improved their interactions with adolescent girls and enabled girls to seek more opportunities in their communities.

Among the gender-related attitudes and norms discussed in the interviews, masculinity, gender roles, girls’ education, and early marriage were most prevalent. Adolescent boys, male caregivers, and female caregivers consistently reflected on the ways in which conceptions of masculinity and manhood impacted their relationships with their sisters and daughters, and many participants reflected on how these conceptions inhibited healthy interaction and connection within the household. In particular, interrogations of masculinity often resulted in improved behavioral control and communication, alongside more equitable decision-making and assignment of roles within the household. When presented with a hypothetical scenario, adolescent boys discussed changes in their perceptions of masculinity, noting that they previously included the use of physical violence as integral to manhood: “He hit his sister and thought he was a man” (Azraq-PAR-AB-15022022E). Adolescent boys noted that, through the program, they began to reconceptualize masculinity as inclusive of equitable distributions of household chores and support of siblings. Female caregivers also noticed the impact of these altered definitions of masculinity on gender roles and on providing support and guidance: “I’ve seen that the relationship between my son and daughter has changed after the program…He thought that he was the guy and she was the girl and end of story. The program helped him change this idea and helped us become cooperative and participative” (Azraq-IDI-CF-14022022(1)(E)).

In addition to masculinity, participants also examined their preconceived notions of a girl’s right to education and its relation to early marriage. More specifically, the SSAGE program appeared to prompt constructive conversations amongst participants about community level norms, such as child, early, and forced marriage (CEFM) and removing girls from school. One program mentor discussed the ways in which paternal caregivers reflected on these issues: “There was a father who made his daughter drop out of school in the sixth grade. This was six years ago. In our first course, they discussed these issues under social violence. The man said that he regretted making her leave school in the sixth grade, and that he would let her go back to school if he could” (KII02). These changed attitudes in turn seemed to spur shifts in participants’ behaviors. Regarding the individual-level changes they observed, one program mentor noted, “Early marriage happens in the camp and not all people let their daughters finish their education. Through the sessions, some dads allowed their daughters to go back to school and to live their lives normally, to choose what she wants and not to be forced to marry. They changed some of their ideas” (KII01). Adolescent girls also spoke to increased access to education, noting, “Our parents didn’t let us go to school but now they let us. After the course they allow us…He said that this school doesn’t teach a thing and that I would get an education when we go back to Syria. Now he tells me to go to school and not to quit it and to learn” (Azraq-PI-AG-14022022(1)(E)).

SSAGE further fostered reflection on the importance of equal access to education, which allowed for greater connection, empathy, and understanding among siblings: “A brother can understand his sister now. If she goes out he will not prevent her or ask her why. School is school. He knows that she has the right to learn as he does. He should also make sure that she gets education if he himself can’t” (Azraq-FGD-CF-15022022(E)). Another male caregiver considered his change in opinion regarding girls’ education, which tangibly impacted his daughter’s life: “I’m one of the people who was against a girl’s education…not anymore” (Za’atari-IDI-CM-13022022(3)(E)(1)). He continued by sharing that prior to the program, he had stopped her education, but after, he plans to allow her to attend school.

Adolescent girls’ well-being was also affected by a shift in individual conceptions of gender roles. Participants often discussed positive changes in household structure, including challenging the gendered order of domestic responsibilities and normalizing male participation in household chores. One female caregiver discussed how the sole burden of household responsibilities no longer falls on her adolescent daughter and herself, and is rather shared among the family:


*“My husband and I for example… he didn’t attend the sessions at first, then he did. The boy used to tell the girl to wash his dishes, and their dad used to tell me to wash the dishes. My son used to tell his sister to prepare the bed for him. After my son participated in the sessions, he started to prepare his bed and wash his dishes. He changed the way he treated his sisters. He is serving himself now. He learnt about compassion and when he sees that his sister or I are sick he will serve himself. His father used to say: ‘is he the girl of the house?’ and that we should serve him. Now things have changed”.*
(Za’atari-FGD-CF-24102021(2))

Despite these positive changes, it is important to note that some female caregivers suggested that the SSAGE program reduced gendered inequity in household labor between their sons and daughters to varying degrees. While some brothers took over their sisters’ housework to actively equalize labor sharing, others merely stopped expecting their sisters to fulfill tasks in service of their needs “…when he sees that his sister or I are sick he will serve himself. His father tells him to assume that his sister is not there and that she doesn’t have to do everything” (Za’atari_FGD_CF_24102021_2).

#### 3.2.3. Changes in Control of and Communication around Girls’ Mobility and Behaviors

Study participants described values and practices within their camp communities, including those that restrict adolescent girls’ mobility in the name of safeguarding their sexual purity and, consequently, their families’ honor. In addition to caregivers’ changed attitudes toward and permission for girls’ schooling, participants reported some shifts related to other behavioral control practices, including restrictions on adolescent girls’ mobility outside the house and demands for girls to carry out some tasks within the household. One maternal caregiver shared that her son started to approve of his sister traveling outside the house for shopping and schooling: “Her brother used to dominate her. Now things are better between them. ‘Go beloved sister, make us some tea. What do you want for dessert? Go with mom and buy it.’ Before, she wasn’t allowed to go out. He asks her about her school now. He never did that before” (Za’atari-IDI-CF-24102021(1)). Similarly, an adolescent boy shared that after the program, he too changed in his control of his sisters’ mobility in the community. Instead of forbidding her from leaving the household to visit a friend, he now accompanied his sister to the friend’s household: “[The SSAGE Program] made us aware of things we didn’t realize before. For example, when my sister wanted to visit her friend, I used to tell her to stay home before the program. After the program, I take her there” (P5, Azraq-PAR-AB-17102021(1)).

While this action demonstrates some improvement in the girl’s ability to move freely about her community, it is important to note that the male sibling continues to supervise her movements, and that remains normalized in the eyes of some caregivers and adolescents. Nonetheless, because of adolescent boys’ enhanced understanding of protection and safety of adolescent girls, one adolescent boy stopped punishing his sister for experiencing harassment from others. According to his sister, he instead trusts and protects her: “Sometimes, a girl faces a trouble in the street. A guy follows her, and someone sees that. The brother knows and he doesn’t trust his sister anymore. He suspects her and she doesn’t leave the house without him. The program teaches him that if something like this happened, he shouldn’t suspect his sister. He should trust her and answer back people who talk about her” (Za’atari-PAR-AG-20102021(1)).

#### 3.2.4. Affective Involvement and Emotional Connection within the Household

According to study participants, the SSAGE program fostered affective involvement between caregivers and brothers and adolescent girls, ultimately increasing girls’ sense of self-confidence and self-efficacy, improving their psychological well-being and encouraging prosocial engagement with their family and peers. With improved closeness, warmth, and communication, adolescent boys exhibited elevated regard for their sisters’ emotions, concerns, and psychosocial needs. As described by one adolescent girl participant: “If he sees me upset, he will calm me and ask me what’s wrong. He tells me things about his life” (Azraq-PI-AG-14022022(1)(e)). Such sympathetic sibling relationships induced by the SSAGE programming encouraged girls to feel comfortable enough to ask for help and confide in family members when facing trouble in and out of the home. One adolescent male shared his experience with his sister, saying that after SSAGE, “If she needed help with homework, I would help her. I changed after the training. If she faced a trouble inside the house, she would tell her mother. If she faced trouble outside the house, such as being followed by a guy, she would tell her father or her big brother” (Azraq-PAR-AB-17102021(1)).

Cultivating greater affective involvement between mothers and daughters improved adolescent girls’ well-being in various domains—particularly prosocial behavior, self-confidence, and mental health. According to many participants, girls shared their experiences and exhibited greater familial interaction when they perceived greater interest in communication and engagement from their mothers. One mother described the change in dynamics, stating, “My daughter used to come to me to tell me things she’s gone through. I used to tell here that I was busy and ask her to leave. I used to listen a little and keep telling her to finish quickly. Now I listen more and give her time. She talks to me about school and things she faces there, and I answer her” (Azraq-IDI-CF-18102021(3)). Respondent interviews also exhibited a link between this pattern of warmer parenting and girls’ emotional well-being, suggesting an association not only between the increase in mothers’ affective involvement and adolescent girls’ prosocial behavior, but their mental health, as well. As illustrated by a female caregiver:


*“I did not sit with my daughters a lot before. Now I sit with them and talk to them. I feel that a mother should do this. A girl will not tell her mother things if she feels that her mother is distant. When a mother designates time to sit with her daughters, they will communicate more and open up to one another…Yes, there is more harmony now. Anger and sadness are gone. For example, my daughter is not upset with me anymore”.*
(Azraq-IDI-CF-18102021(1))

Another key theme throughout the interviews was the effect of increased emotional connection between mothers and daughters on greater expression of self-confidence and perceived safety among adolescent girls. When girls felt valued and had their perspectives considered and validated, they were observed exhibiting improved security and self-assurance in voicing their thoughts. As recounted by one mother, “We benefited a lot I swear…A mother understands her daughter and asks for her opinion. A daughter feels more comfortable talking to her mom. She feels safe and confident to speak up and tell what she went through” (Azraq-FGD-CF-15022022(e)).

## 4. Discussion

This study presents findings from a mixed methods pilot evaluation of the SSAGE intervention in the Azraq and Za’atari camps in northern Jordan and highlights the program’s potential to improve adolescent girls’ well-being in humanitarian settings. Quantitative findings revealed improvements in adolescent girls’ mental distress and resilience and, qualitatively, positive changes in girls’ well-being were often discussed by participants as including increased self-confidence and agency and improved prosocial behaviors, both with peers as well as family members. Further analysis suggests the improvements in well-being for adolescent girls were facilitated through multiple pathways, contingent on participation from all four cohorts, including participants’ reflection on and reconceptualization of gender dynamics and inequity, particularly with respect to gender roles and girls’ education, a shift toward healthier communication between girls and their household members, and greater emotional connectivity and support from family members for girls.

Despite the positive impacts resulting from improved family functioning, increased awareness around girls’ safety, and changes in family members’ gender equitable attitudes and behaviors, it is important to note that participants largely conceptualized improvements to girls’ well-being and freedoms within a set of norms that ultimately remained restrictive. For example, while many participants noted that adolescent girls’ mobility improved following the program, this change was often a result of brothers being more amenable to accompanying their sisters outside of the household, as brothers now recognized that mobility supported their sisters’ well-being. The data do not show shifts in community-level norms, which dictate, for example, that girls need be accompanied by male family members when moving through the public space. This lack of community-level change is unsurprising, as the SSAGE program was implemented over the course of three-month intervals with a relatively small group of people and focused primarily on individual attitudes, behaviors, and interpersonal relationships within family or household units. However, it is important to note that even though community-level norms were not addressed via SSAGE, the positive impacts to girls’ well-being resulting from changes in their own self-perception as well as the behaviors and attitudes of their direct familial support network help to increase girls’ resiliency and reduce the risks they face living in contexts where norms which restrict their freedoms remain prevalent. Nonetheless, future iterations of SSAGE might consider adding on complementary programming at the community level to raise awareness and challenge gender inequitable norms.

While adolescent girls’ own participation in the SSAGE program sessions appears to have conferred direct benefits—for example, our findings reveal girls’ increased awareness around bullying and a reduction in related anti-social behavior—several improvements in girls’ well-being and mental health seem to be supported indirectly through the participation of their family members. Although biological differences explain at least some of the disproportionately higher rates of depression in adolescent girls as compared to boys, Cyranowski and colleagues [12] posit that girls also face a greater risk of mental health disorders due to intensified gender socialization and impaired adolescent transitions facilitated by (i) insecure attachment styles, (ii) anxious or inhibited temperament, and (iii) low instrumental coping skills. Findings from this study suggest that girls’ participation in their peer group sessions contributed to greater self-confidence, self-efficacy, and healthy coping mechanisms, as demonstrated by seeking family support. Additionally, findings point to the influence of girls’ family members’ participation on overall family attachment and support, as well as self-efficacy. We propose a conceptual framework of these pathways in Figure 2.

Findings from this evaluative research demonstrate that the family-based approach and its engagement of male and female caregivers and, importantly, male siblings, support improved empathy towards girls, strengthen intrahousehold communication, and promote a healthier and more equitable family dynamic, all of which proved to be relevant for girls’ well-being. Evaluations of programs aimed at improving the well-being and safety of adolescent girls in humanitarian settings have concluded that working with girls and female caregivers is not sufficient to induce positive changes around safety and gender equity [45]; these studies argue that programs must also engage male caregivers and other men in girls’ lives who hold decision-making power in these contexts. A recent study in the Democratic Republic of Congo found positive associations between men’s family functioning and positive parenting and power-sharing, further emphasizing the value in engaging male caregivers in programs for adolescent girls [46,47].

In addition to engaging male caregivers, brothers, and other men in girls’ lives, our findings reiterate the importance of employing a gender transformative approach with all program participants. A previous qualitative evaluation of SSAGE in Borno State, Nigeria found that improvements in family functioning along gender dimensions, specifically, seemed to leverage the most positive change for adolescent girls [33]. Previous studies from humanitarian contexts have also found associations between improvements in caregivers’ gender equitable attitudes and school attendance for girls as well as correlations between perceived gender norms within a girl’s family and her self-esteem [15,48]. As such, findings from this evaluation point to the cruciality of both the whole-family and gender-transformative dimensions of SSAGE in supporting positive changes for adolescent girls.

Findings from this study should be considered alongside a few limitations. First, due to resource constraints, the quantitative survey questionnaire was administered to a small sample. Future quantitative evaluations of SSAGE should employ a larger sample size, where logistically feasible, in order to identify the impacts of the intervention with greater statistical power. Second, it is important to recognize the self-reported nature of the quantitative outcomes of interest, which may be subject to self-reporting bias. Third, quality control reviews of the qualitative transcripts against the audio files revealed some minor issues with transcription quality, which may have impacted the transcripts’ accurate representation of participants’ full thoughts and opinions. In future qualitative evaluations of SSAGE, it is recommended that transcription standards and practices be reinforced prior to the start of the transcription and more regular spot checks comparing transcripts to audio files be conducted during the transcription process to ensure quality. Finally, it is important to acknowledge the limitations of collecting data immediately following the intervention. Gender norms are complex and deeply embedded social constructs; changing these norms, particularly at the community level, is a substantial undertaking and can take several months or even years. As endline data were collected immediately following the end of the intervention, it is unlikely that the data would have revealed shifts in community-level norms. Longer term follow-ups are needed to understand whether the program induced changes in norms as well as to determine whether observed improvements at endline are sustainable in the longer term.

## 5. Conclusions

Despite these limitations, findings from this mixed-methods evaluation point to the potential of the SSAGE intervention for supporting the well-being and mental health of adolescent girls in humanitarian settings. Data from this study highlight changes in gender equitable attitudes, improved communication within the household, and strengthened emotional connection and involvement between family members as pathways that supported adolescent girls and indirectly led to their improved mental well-being. Practitioners in humanitarian settings might usefully integrate gender-transformative and whole-family approaches in programming in order to maximize positive impacts for adolescent girls.

## Figures and Tables

**Figure 1 ijerph-19-15357-f001:**
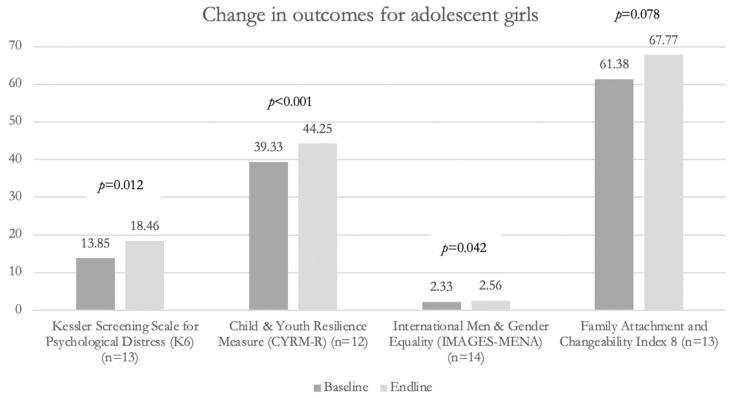
Change in outcomes for adolescent girls only. Note: Baseline and endline data were collected within two weeks immediately before and after the SSAGE intervention, respectively.

**Figure 2 ijerph-19-15357-f002:**
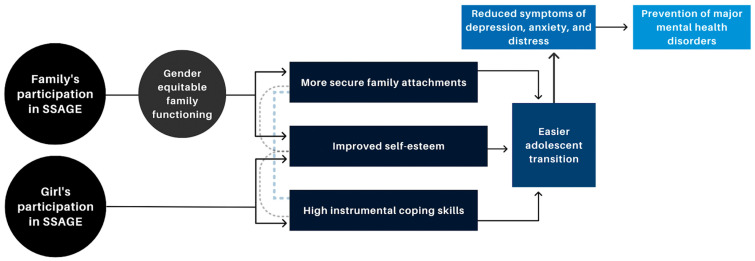
Conceptual framework of SSAGE’s impact on adolescent girls’ well-being and mental health.

**Table 1 ijerph-19-15357-t001:** Qualitative methods sample.

		Adolescent Girls	Adolescent Boys	Female Caregivers	Male Caregivers	SSAGE Program Staff and Mentors
Cycle 1	Azraq	19	16	12	15	–
	Za’atari	20	19	17	19	–
Cycle 2	Azraq	17	16	12	12	–
	Za’atari	9	4	3	3	–
Total		65	55	44	49	8

**Table 2 ijerph-19-15357-t002:** Descriptive statistics.

	Full Sample (n = 68)	Adolescent Girls (n = 18)	Adolescent Boys (n = 17)	Female Caregivers (n = 16)	Male Caregivers (n = 17)
Age	29.7 [16.2]	12.5 [2.0]	16.7 [2.7]	41.2 [4.0]	48.6 [7.6]
Ever attended school	66 (97.1%)	18 (100.0%)	16 (94.1%)	15 (93.8%)	17 (100.0%)
Currently attending school (if ever attended)	---	18 (100.0%)	12 (92.3%)	---	---
Ever worked for pay	26 (38.2%)	1 (5.6%)	1 (4.2%)	4 (25.0%)	14 (82.4%)

All statistics presented are number of participants and percentages, except for age, which is summarized as a mean and standard deviation (in brackets).

**Table 3 ijerph-19-15357-t003:** Change in outcomes for full panel sample.

	Baseline	Endline	*p*-Value
Mental Health			
Kessler Screening Scale for Psychological Distress (n = 55)	13.67	15.85	0.007
	[5.16]	[6.04]	
Resilience			
Child and Youth Resilience Measure (n = 24)	39.83	42.62	0.030
	[4.78]	[5.06]	
Gender Equity			
International Men and Gender Equality-modified scale (n = 58)	2.34	2.49	0.004
	[0.48]	[0.43]	
Family Functioning			
Family Attachment and Changeability Index (n = 52)	61.33	64.69	0.044
	[8.30]	[10.01]	

Note: Baseline and endline data were collected within two weeks immediately before and after the SSAGE intervention, respectively. Differences between outcomes at baseline and endline were assessed using paired *t*-tests.

## Data Availability

Not applicable.

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
