# Peer review of "Exploring the Impact of a Family-Focused, Gender-Transformative Intervention on Adolescent Girls’ Well-Being in a Humanitarian Context"

_ijerph, 2022, doi:10.3390/ijerph192215357_

Round 1
Reviewer 1 Report
I enjoyed reading the article. Overall, it is well-written and its arguments justified.
However, I have a few comments that might help strengthen the article:
1. Introduction
· Check in-text referencing;
· I believe that the concepts of culture, gender norms, honour and patriarchy should be further discussed. From my interpretation, the authors take these concepts for granted, without critically engaging with them. In my reading of the article, young women are presented as victims/passive/without agency previous to the intervention, whereas young men are presented as perpetrators/controlling/uncaring. Gender issues in the MENA region are not discussed and culture seems to be presented as static. The concept of patriarchy is also not defined. While I am by no means diminishing the importance of underscoring unequal and harmful gender (power) relations, I believe that it is important to present these issues in a more complex and dynamic way that takes into account intersecting social positions.
· The impact of conflict and economic issues on gender (power) relations in (post)conflict/humanitarian contexts are mentioned very briefly. Despite this not being the main focus of the study, the authors could further highlight its (potential) importance/impact on gender (power) relations within the family.
2. Methodology
· Is there a link/reference for the ISHRAK project?
· “(On a 5-point Likert scale, from ‘Not at all’ to 190 ‘A lot’)” (p. 4) – “not at all” and “a lot” are subjective measurements; that is, they might be interpreted differently. The authors could acknowledge the potential subjective interpretations of the questionnaire.
· Can you say slightly more about the different qualitative methods of data collection used? Did the different qualitative methods of data collection have the same results? Where the social interactions during the focus group taken into consideration? Was there conformity between participants, as can be expected during focus group discussions?
· How many people exactly participated in the qualitative research methods data collection? It is interesting that there where 68 participants in the questionnaire, but it is mentioned that for the for qualitative research methods of data collection: “In total, 137 [participants] were selected from the first program cycle and 76 were from the second program cycle” (p. 5). Are participants in the first and second cycles the same? (If it is the case) how is it that there were more participants for the qualitative research methods of data collection than the quantitative ones?
3. Results
· What do traditional cultural values relate to? (p. 12) These should be discussed in the literature review. See previous comments.
4. Discussion
· “Although biological differences explain at least some of the disproportionally higher rates of depression in adolescent girls as compared to boys” – shouldn’t this be discussed in the literature review?
5. Limitation (p. 15/16)
· The authors recognizes that gender norms are complex, but they seem to be taken for granted and not discussed enough in the literature.
Reading suggestions:
Joseph, Suad, editor. Arab Family Studies: Critical Reviews. Syracuse University Press, 2018. JSTOR, https://doi.org/10.2307/j.ctt1pk860c. Accessed 17 Oct. 2022.
Kandiyoti, D. (1988). Bargaining with patriarchy. Gender and Society, vol. 2(3), pp. 274-290. https://www.jstor.org/stable/190357
Tripathy J. (2010) ‘How Gendered Is Gender and Development? Culture, Masculinity, and Gender Difference’. Development in Practice 20(1): 113-121.
Walby, S. (1989). Theorising patriarchy. Sociology, 23(2), 213-234. https://doi.org/10.1177/0038038589023002004
Reviewer 2 Report
-I recommend modifying the title, the first part partially represents the topic addressed
-There are flaws in the way of citing APA, especially in the first pages. For example, in line 37 both authors go inside the same parentheses (Blackmore et al., 2020; Baauw et al., 2019; World...). Also, the dot always goes at the end of the parentheses, and not before it. This mistake is made several times throughout the text.
-It is necessary to explain in detail what it consists of the SSAGE program in order to understand the results
-Why did the researchers take a month as a criterion to check changes after the program?
-Authors should include the psychometric properties of the questionnaires used
-Table 3 should be presented disaggregated (girls, boys, female caregivers and male caregivers)
-Statistical analyzes are very basic, limited to presenting descriptive statistics and paired t-tests.
-It would be interesting if they included examples in the qualitative part of speeches in which there has been no change.
-An important limitation is that there were no changes at the community level. This can cause the effects of the program to wear off after a while.
Round 2
Reviewer 2 Report
The authors have addressed all the questions raised